# Effects of Ply Orientations and Stacking Sequences on Impact Response of Pineapple Leaf Fibre (PALF)/Carbon Hybrid Laminate Composites

**DOI:** 10.3390/ma15176121

**Published:** 2022-09-03

**Authors:** Mohd Khairul Rabani Hashim, Mohd Shukry Abdul Majid, Mohd Ridzuan Mohd Jamir, Farizul Hafiz Kasim, Hassan A. Alshahrani, Mohd Azaman Md Deros, David Hui

**Affiliations:** 1Faculty of Mechanical Engineering Technology, Universiti Malaysia Perlis (UniMAP), Arau 02600, Perlis, Malaysia; 2Faculty of Chemical Engineering Technology, Universiti Malaysia Perlis (UniMAP), Arau 02600, Perlis, Malaysia; 3Department of Mechanical Engineering, College of Engineering, Najran University, Najran 61441, Saudi Arabia; 4Department of Mechanical Engineering, University of New Orleans, New Orleans, LA 70148, USA

**Keywords:** hybrid composite, impact response, PALF, ply orientation, stacking sequences

## Abstract

This study investigated the impact response behaviours of pineapple leaf fibre (PALF)/carbon hybrid laminate composites for different ply orientations and stacking sequences. The laminates were manufactured using a vacuum infusion approach with various stacking sequences and ply orientations classified as symmetric quasi-isotropic, angle-ply symmetric, and cross-ply symmetric. The laminates were analysed using an IMATEK IM10 drop weight impact tester with an increment of 5 J until the samples were perforated. This investigation reveals that the overall impact properties of PALF and carbon as reinforcements were improved by a beneficial hybridised effect. The laminates with an exterior carbon layer can withstand high impact energy levels up to 27.5 J. The laminate with different stacking sequences had a lower energy transfer rate and ruptured at higher impact energy. The laminates with ply orientations of [0°/90°] and [±45°]_8_ exhibited 10% to 30% better energy absorption than those with ply orientations of [±45°_2_, 0°/90°_2_]_s_ and [0°/90°_2_, ±45°_2_]_s_ due to energy being readily transferred within the same linear ply orientation. Through visual inspection, delamination was observed to occur at the interfaces of different stacking sequences and ply orientations.

## 1. Introduction

Carbon-fibre-reinforced polymer (CFRPs) composites are widely used in modern industries because of their superior mechanical strength [1]. However, CFRPs are harmful to the environment to a certain extent. Therefore, a hybrid synthetic with natural fibres was introduced to ensure environmental sustainability and reduce the carbon footprint. Hybrid fibre-reinforced composites comprise at least one pair of two different fibres combined in a solitary polymer matrix, resulting in improved properties compared to a normal polymer composite. There are several different definitions of hybrid composites provided by different researchers. Thwe et al. [2] described hybrid composites as reinforcing materials that combine multiple reinforcement fibres or matrices (blends) to provide strength and durability. Furthermore, they can be incorporated into two or more reinforcing and filling materials in a single matrix of reinforcement and filling materials [3]. Hybrid composites are widely used in real-life applications [4,5,6,7]. Hybrid composites are more advanced than conventional fibre-reinforced composites and have more potential for application than other composite materials. Natural-synthetic fibre hybrid composites have been the subject of previous research, which mostly focused on reducing the use of synthetic fibres [8,9]. A previous study discussed the potential benefits of natural-synthetic fibre hybridisation and its implementation [10]. One of the best options for natural reinforcement fibre is pineapple leaf fibre (PALF), a massive amount of biomass waste abundantly available in tropical countries [11]. Furthermore, among the several natural fibres extracted from plant leaves, PALF has the largest portion of fibre content and the minimum microfibrillar angle, which is the primary reason for its excellent impact performance [12]. PALF is the most often used fibre in the textile industry for various reasons, including its abundance, low cost, superior thermal and acoustic insulation, exceptional tensile strength, and high toughness.

In agriculture, pineapple fruit is considered a primary crop since it is grown for human use, but pineapple leaves are regarded as a secondary crop or trash. A valuable agricultural waste widely accessible in tropical nations, particularly Malaysia, is the pineapple leaf. The leaf produces a lot of cellulose fibre, with cellulose making up the majority (70–82%) and lignin (5–12%) and ash making up the remainder (1.1%) [13]. Because of their excellent mechanical strength and low cost, PALF has high application potential as biodegradable plastic composites [14], reinforced polymer composites [15,16], low-density polyethylene (LDPE) composites [17], thermoset composites [15], thermoplastic composites [18], and rubber composites [19]. By adjusting the matrix ratio, fibre length, stacking order, and fibre orientation, PALF/carbon fibre hybrid composites may be created to satisfy specific applications and achieve different mechanical and physical qualities [20,21].

Reinforcement fibres are the primary load-bearing components, accounting for the majority of the strength and stiffness of the composite. Ply orientation is important in designing composite laminates to withstand high-impact loads. Ply orientation in composites has a complex relationship with their impact damage resistance because of the multidirectional behaviour of the composite and the mechanism through which the damage propagates through the laminate. The quality and strength of adhesion (bonding) within the fibre/matrix system are important components of the resistance of the composites to impact damage [22]. In a previous study, in comparison to the other investigated fibre orientations of [0°/+60°/−60°]_s_ and [0/+45°/−45°]_s_, Belingardi and Vadori [23] discovered that glass fibre composites with a stacking sequence of [0/90°] had the highest saturation energy and the best impact resistance. Another investigation was performed by Sikarwar et al. [24] on the impact response of woven glass fibre composites as a function of thickness and fibre orientation. According to their findings, [0/90°] laminates exhibited the highest impact resistance across all the examined lay-ups, mainly due to the failure strain, which is highly influenced by the fibre orientation in the laminate. Quaresimin et al. [25] observed that the impact energy absorption capabilities may be affected by the thickness and fibre orientation, with a [0°/45°] interface demonstrating the least impact damage.

The stacking layering sequence in the composite structure is another factor that influences the impact strength in addition to ply orientation. Researchers studied the reactions of four different hybrid laminates under low-velocity impact loading [26,27]. The results demonstrated that the load-carrying capabilities of hybrid composites are significantly improved than carbon/epoxy laminates, with only a minor compromise in stiffness. The impact behaviour of hybrid composite plates was examined by Sayer et al. [26]. Two types of hybrid composite plates (glass–carbon/epoxy) were subjected to impact tests until they were completely perforated. The load–deflection curves and photographs of the damaged samples acquired from the impacted and non-impacted sides were compared to determine the failure processes of the damaged specimens for various impact energies. The perforation threshold of the hybrid composite with a carbon face sheet was 30% more than that of the hybrid composite with a glass face sheet. The low-velocity impact testing by Sarasini et al. [27] investigated the damage tolerance of carbon/flax composites to determine their failure. All the samples with different stacking orders were tested for their flexural strength. In comparison to the samples coated with natural fibre layers, the samples covered with high-strength carbon layers exhibited higher flexural strength at low impacts.

However, the flexural modulus was based solely on the specimens’ volume percentage of carbon fibre. The results were comparable to those of other polymer composites reinforced with natural and synthetic fibres [28]. In addition, Selver et al. reported the impact and post-impact behaviour of glass fibre-reinforced polymer composite (GFRP) laminates and hybrid glass/natural fibre-reinforced polymer composite laminates constructed using various layering sequences [29]. The impact resistance of the GFRP laminates was higher than that of hybrid laminates. Hybrid laminates with glass fabric in the outer skin (skin) exhibited higher impact strength than laminates with glass fabric in the core (core). In the same investigation, natural fibre and hybrid composite laminates absorbed more energy than the GFRP composite laminates. Consequently, the effects of the glass/natural fibre-reinforcement stacking sequence were also examined in various hybrid formulations with GFRP [30,31,32].

In previous studies, complete investigations of PALF as reinforcement for polymer composites and the mechanical characteristics of the hybrid laminate composite were performed [15,33,34,35]. However, the effects of ply orientation and stacking sequences on the low-energy impact of PALF/carbon hybrid laminate composites are yet to be investigated. Therefore, detailed analyses of the low-velocity impact behaviour of PALF/carbon hybrid laminate composites were performed in this study. In order to comprehensively understand the behaviour of the composites after impact, the interplay between ply orientation and stacking sequences is required. Due to the high potential of the PALF/carbon laminate composite, it is possible to explore its use in future applications.

## 2. Materials and Methodology

### 2.1. Materials

The laminates were fabricated using plain-weave PALF (185 GSM), twill weave carbon fibre mat (200 GSM), EpoxAmite 100 epoxy resin (epoxy polymer (ether of bisphenol A)), and a hardener (triethylenetetramine). The raw materials were supplied locally by Mecha Solve Engineering Sdn. Bhd, Kuala Lumpur, Malaysia. The specifications of the raw materials used are listed in Table 1. Alkaline treatment was performed on PALF before fabrication with a soaking time of 3 h in a 5% alkali (NaOH) solution at a 40:1 liquor ratio [36]. The treated PALF was dried for 8 h at 60 °C and thereafter dried further for 24 h at room temperature.

### 2.2. Specimen Preparation

The specimens were manufactured using a vacuum-infusion technique. The PALF and carbon plies were layered on the glass surface and protected by a plastic layer. The resin was injected into the lamination plies using a high vacuum pump (AST 22, AIRSPEC, Kuala Lumpur, Malaysia). The initial curing step took place in the mould for 12 h at ambient temperature, followed by 2 h post-curing in an oven at 80 °C with air circulation. The layering sequences of the laminates are summarised and illustrated in Table 2a,b.

### 2.3. Experimental Procedure

The impact responses of the laminates were evaluated using an IMATEK IM10, Herts, UK, instrumented drop-weight impact tester in compliance with ASTM D7136-15. The samples had dimensions of 150 mm × 100 mm × 5 mm. The machine was outfitted with a 9.68 kg impactor and a hemispherical impact head with a diameter of 10 mm and a mass of 0.71 kg. As illustrated in Figure 1, the laminate was clamped between two metal features with a central circular aperture where the impact occurred. The laminates were tested at varying energy levels with increments of 5 J until the samples were perforated. In order to determine the specific perforated energy levels, a refinement test was performed between the unperforated and perforated energy levels. Five replicates were tested for each laminate layer sequence. Finally, the peak load and deflection experienced by the specimens were collected and computed using a data acquisition system. A post-analysis was performed using MATLAB R2021a to compute energy absorption.

## 3. Results and Discussion

### 3.1. Impact Behavior

The purpose of impact testing was to determine the influence of ply orientation and stacking sequence on the behaviour of the PALF/carbon hybrid laminate composite. The energy levels applied to the laminates are listed (Table 3).

The collected raw data demonstrate an oscillating and noisy behaviour response to the natural modes of vibration of the impacting system; shaft, hammer, and impact sensor [40]. In order to estimate the trend curve of the contact force response, the data were presented in moving average smoothing form. The satirical technique was established in previous studies [28,41]. Figure 2 shows an example of the contact force–displacement curves of the CPPC laminate with a ply orientation of [±45°]_n_ at an energy level of 10 J. The moving average data were computed based on 50 sampling data ranges, known as 50 MA. The 50 MA is the most advanced of the three averages; therefore, it serves as the first line of major moving average support in an uptrend or the first line of major moving average resistance in a downward trend. The slope of the curve represents the laminate stiffness. 

On average, the hybrid laminates exhibited superior impact strength over the non-hybrid laminates. The impact behaviours of the hybrid laminates at varying ply orientations and stacking sequences are shown in Figure 3, Figure 4, Figure 5, Figure 6, Figure 7, Figure 8, Figure 9 and Figure 10. The curves generally show the energy levels at which rebounding, penetration, and perforation occur during the impact test. Figure 3 shows the force–displacement curve for the hybrid laminates at a ply orientation of [0°/90°]_8_, while Figure 4 shows the force–time curves. Both the PPPP-untreated and PPPP-treated laminates exhibited the same force–displacement curve trend, starting with a typical rebounding curve at low impact energy until the maximum contact force increased to 12.5 J before perforation at 15 J. The typical rebounding effect was defined by typical load increase and discharge decrease phases with a single-peak load that is often observed in lower impact energy scenarios [42]. The exterior top surface of the PALF layer experiences local bending and indentation in the surrounding impact region; however, the inner layer experiences a localised buckling contact force that reaches the peak load zone. Consequently, the laminate absorbs a modest quantity of irreversible energy. The PPPP-treated sample exhibited a significantly higher maximum contact force than the PPPP-untreated sample, 35% more than the contact force at 2.7 kN, as shown in Figure 5. In addition, the PPPP-treated sample exhibited approximately 30% less displacement indention than the PPPP-untreated sample. The stiffness values of the PPPP-untreated and PPPP-treated samples were significantly different, at which PPPP-treated were able to almost withstand 20% more impact content force. For the PCCP laminate, the curves exhibited a rebound pattern at low impact energy. A maximum contact force of 5.2 kN was determined at 20 J, and perforation occurred at 22.5 J. The CPPC laminate exhibited a similar rebound curve trend up to the maximum contact force. Even with a similar trend, the curve displacement is identically different due to the varying stacking sequences. The CPPC laminate exhibited the maximum contact force at 4 mm; the impact shaft only reacted at the bottom internal layer of the PALF ply before the carbon ply, whereas the PCCP laminate exhibited the maximum contact force at 6 mm (it almost fully penetrated). The CPPC laminate exhibited penetration at 25 J before perforation at 27.5 J. Stiffness of the CPPC laminate was significantly higher than that of the PCCP laminate, at which CPPC laminate was able to almost withstand 30% more impact content force.

Figure 6, Figure 7 and Figure 8 illustrate the impact behaviour of the hybrid laminates at a ply orientation of [±45°]_8_. In Figure 6, the PPPP-untreated and PPPP-treated laminates exhibited a similar force–displacement curve trend. At 5, 10, and 15 J impact energy, the laminates exhibited a rebound effect after the impactor contact force reached the peak force zone. The penetration occurred at 17.5 J before perforation at 20 J. The PPPP-treated laminate revealed approximately 50% contact force higher than the PPPP-untreated, as shown in Figure 8. In comparison to Figure 5, the PPPP-untreated and PPPP-treated were stiffer at [±45°]_8_ ply orientation compared to [0°, 90°]_8_. However, this phenomenon exhibited by the PPPP-untreated and PPPP-treated laminates, which oriented at [0°, 90°]_8_, produced higher tensile strength compared to laminates with [±45°]_8_ ply orientation [43]. The theoretical prediction explained the phenomenon that the laminate with [±45°] ply orientation had better transverse shear strain than that with [0°, 90°] ply orientation, which is useful to withstand the impact load [44]. For the PCCP laminate, the curves exhibited a rebound pattern at low impact energies of 5, 10, and 15 J. The maximum contact was recorded at 4.1 kN with 17.5 J impact energy. The laminate was fully penetrated and perforated at 20 J. The CPPC laminate displayed a high peak load even at 5 J impact energy. The laminates withstand high impact energy before penetration occurs at 25 J, followed by perforation at 27.5 J. The highest peak load for the CPPC laminate was 5.0 kN at 25 J impact energy. By comparing the PCCP and CPPC laminates, the CPPC laminate exhibited higher stiffness compared to the PCCP laminate.

The contact force–displacement curves for the laminates with ply orientations of [±45°_2_, 0°/90°_2_]_s_ are shown in Figure 9. Figure 10 shows the force–time curves. The laminates with ply orientations of [±45°_n_, 0°/90°_n_]_s_ exhibited force–displacement curves that revealed mixed tendencies. This is due to the hybridisation effect of the ply orientation that results in varying stiffness along with the thickness of the laminate [45]. The PPPP-untreated laminate exhibited a rebound effect at 5 and 10 J impact energy levels, followed by penetration at 12.5 J with a maximum curve peak of 2.0 kN, as shown in Figure 11. The laminate was perforated at an energy of 15 J. The PPPP-treated laminate exhibited a slight increase in impact strength as the laminate withstood the rebound effect zone up to 15 J before penetration at 17.5 J with a maximum force peak at 3.1 kN. The overall impact strength increased by approximately double that of the PPPP-treated laminate by incorporating the interior carbon ply in the PCCP laminate. The laminate was penetrated only when the impact energy reached 17.5 J and was perforated at 20 J. The CPPC laminate exhibited the greatest impact strength by referring to the prolonged curve on the rebound zone until penetration occurred at 25 J. The CPPC laminate is able to withstand the highest contact of 6.2 kN compared to other laminates and exhibited a unique curve feature of peak load followed by a prolonged loading plateau, particularly for the impact energy greater than 15 J. Similar observations were reported by He et al. [42]. The fracture of the laminate was initiated from the top exterior carbon layer, as shown by the contact force reaching the peak value. Thereafter, the cracks spread along and perpendicular to the entire surface of carbon ply as the impactor moved downwards; as a consequence, force indentation indicates a prolonged stable plateau right after the peak force. During this stage, indentations and damages were observed in the impact area of the top exterior carbon layer. The interior PALF layer provides sufficient force resistance to hold the laminate and to stop the impactor from penetrating or perforating the laminate at 20 and 25 J, respectively. Furthermore, the external carbon layer was oriented at 45°, which improved the fracture propagation resistance of the laminate [46].

Figure 12, Figure 13 and Figure 14 illustrate the force–displacement and maximum force–displacement curves, respectively, for the laminates with ply orientations of [0°/90°_2_, ±45°_2_]_s_. As shown in Figure 12, the PPPP-untreated and PPPP-treated laminates exhibited almost the same force–displacement curve. The difference between them was that the PPPP-untreated laminate was penetrated at 12.5 J, whereas the PPPP-treated laminate was penetrated at 17.5 J. The maximum force peak for each impact stage of the PPPP-treated laminate was 25%, which is slightly higher than that of the PPPP-untreated laminate. All the PPPP-treated laminates exhibited better impact strength compared to PPPP-untreated laminates. This phenomenon was explained by removing hemicellulose, lignin, waxes, and other contaminants from pure PALF, which improved the fibre–matrix interaction and resulted in better impact strength [47]. Because of the hydrophilic character of PALF and the hydrophobic nature of the polymer matrix, they have weak contact bonding [48]. Chunhong et al. [49] demonstrated that alkali treatment decreases surface polarity and exposes cellulose, increasing the number of potential reaction sites and contact regions between the PALF and matrix. This observation was in agreement with previous studies on the response to the impact of sisal/epoxy composites [50]. The PCCP laminate exhibited a rebound effect up to 15 J impact energy. The laminate was penetrated and perforated at 17.5 and 20 J, respectively. The maximum force peak for the average stage increased by approximately 40% more than the PPPP-treated laminate. In addition, the penetration and perforation occurred at the same impact energy level because of using the carbon ply as the interior layer. However, the CPPC laminate demonstrated better impact resistance; the laminate was penetrated at 25 J and perforated at 27.5 J. The laminate exhibits a fluctuated force–displacement curve at a high impact energy level. This observation was explained by the shearing effect of the interior PALF layer oriented at ±45° [51,52].

### 3.2. Impact Energy Profile

Figure 15 illustrates the energy profiling diagram of the PALF/carbon hybrid laminate composite. In general, the low-velocity impact of the laminates demonstrated that the stacking sequence had a significant effect on the impact damage response of the laminate, which is in agreement with previous studies [28,53,54]. The absorbed energy of the laminates increased to predetermined impact energy and gradually decreased after reaching the maximum energy value. This explains why the elastic potential energy of the laminates is transformed into impactor kinetic energy, decreasing the absorbed energy [55]. This demonstrates that some energy is dissipated and that not all the energy supplied to the laminate is redirected back to the impactor [56].

Figure 15a shows the energy profiling diagram of the hybrid laminates at a ply orientation of [0°/90°]_8_. The PPPP-untreated sample exhibited approximately 50% energy absorption at low impact energies of 5 and 10 J. The maximum energy absorbed was recorded at 12.5 J with 75% absorption, and the adsorbed energy decreased when the laminate perforated at 15 J. The PPPP-treated laminate exhibited approximately the same energy absorption trend as that of the PPPP-untreated laminate, with a 10–20% improvement. The same trend was reported in previous studies, which concluded only a slight effect on energy absorption after fibre treatment [47]. Even though CPPC could withstand more impact energy than PCCP, it demonstrated a slightly lower absorbed energy. The situation is visible at the maximum impact energy for both laminates, with the PCCP absorbing nearly all of the energy and the CPPC absorbing just approximately 80%. The energy absorbed by the local deformation decreases due to the stiffness of the carbon outer layer, resulting in a more brittle fracture [57].

The energy profiling diagram of the hybrid laminates at a ply orientation of [±45°]_8_ is illustrated in Figure 15b. Although they could sustain a higher impact, all laminates absorbed less impact energy than the laminates at [0°/90°]_8_ ply orientation, particularly the pure PALF laminate. In comparison to the PPPP-untreated at a ply orientation of [±45°]_8_, the hybrid laminate absorbed only 57% of the maximum impact energy at 17.5 J, compared to more than 80% of the PPPP-untreated at a ply orientation of [0°/90°]_8_ with a maximum impact of 12.5 J. A similar trend was observed for the PPPP-treated laminate. This could contribute to the bending stiffness of the laminates; additional elements affect the amount of energy absorbed when the ply orientation changes [58]. PCCP and CPPC laminates absorbed energy well at low impact; however, they degraded significantly at high impact energies.

Figure 15c,d illustrate the energy profiling diagrams of the laminates at ply orientation quasi-isotropic. When the impact energy is delivered at a low level, the computations demonstrate that quasi-isotropic laminates absorb less energy than the cross- and angle-ply laminates. The absorbed energy in quasi-isotropic laminates is equivalent to or slightly higher than that of cross- and angle-ply laminates. This could be because carbon has a higher failure strain than PALF due to its greater elongation. The same absorbed energy trend was reported by Giasin et al. while studying the impact properties of carbon/glass laminates [58]. This phenomenon is further explained by the fact that the energy in a composite laminate can easily be passed from one ply to the next if stacked in the same order [59]. Laminates with varying stacking sequences have a lower energy-transfer rate and rupture when subjected to a higher load [60]. The interlinear interface between laminates with various ply orientations is mechanically weak due to a mismatch in the bending deformations of adjacent plies [61]. The CPPC laminate at [±45°_2_,0°/90°_2_]_s_ ply orientations demonstrated the maximum absorbed energy when it was subjected to the highest impact energy in which almost the entire impact energy was absorbed.

### 3.3. Impact Fracture Morphology

Table 4, Table 5 and Table 6 summarise the results of the investigations on the damage caused by the fracture of the impact. In general, damage to the top surfaces manifests as circular indentations caused by the descending hemispherical head, with the depth of this depression increasing with increasing impact energy. Three types of impact energies were investigated: indentation (maximum), penetration, and perforation. The indentation (maximum) indicates the impact energy level before the laminates were penetrated during the test. The penetration level was determined at the point of the greatest contact force and energy absorption. Table 4 lists the damaged areas exhibited by PPPP-untreated and PPPP-treated at ply orientations of [0°/90°]_8_. The indentation (maximum) for PPPP-untreated occurred at 10 J, which was 5 J less than that of PPPP-treated. The PPPP-untreated exhibited no visible indentation compared to PPPP-treated with a mild circular indentation at the top side and a hairline crack propagated along the ply direction. During this stage, mild external damage was observed because most damage occurs on the internal structure. The internal matrix cracking and delamination was the main damage mechanism at the impact energy levels of rebounding [41]. The penetration energy levels were 12.5 and 17.5 J for the PPPP-untreated and PPPP-treated, respectively. Both exhibited a circular penetration pattern at the top side and a crack opening on the bottom side. This phenomenon explained why the PPPP-untreated was brittle and was proven by the sudden fracture formation at the penetration level compared to the previous indentation (maximum) level. The finding was agreed upon by Romasko in his study on composite oriented at [0°/90°] [62]. In his study, based on c-scan images for front and back sides, the delamination starts around the laminate’s central plane and spreads beneath it, giving the impression that the virtually rectangular zone is a little thinner. The PPPP-untreated perforated at 15 J, whereas the PPPP-treated perforated at 20 J. The PPPP-untreated exhibited a unique fracture pattern with multidirectional cranking propagation. It was demonstrated that the PPPP-untreated polymer matrix could not properly distribute the impact load due to the hydrophobic nature of the polymer matrix, resulting in poor interfacial bonding between them [48].

Table 5 lists the damaged areas of the PCCP and CPPC laminates at ply orientations of [0°/90°]_8_. For the PCCP laminate, indentation (maximum) occurred at 15 J, followed by penetration at 20 J, before perforation at 22.5 J. The CPPC laminate demonstrated a significantly greater impact level at 20, 25, and 27.5 J for indentation (maximum), penetration, and perforation, respectively, compared to the PCCP laminate. As summarised in Table 5, crack propagation was substantially slower in laminates with exterior carbon layers than in laminates with PALF exterior layers. Compression—and tension—shear failures are possible as a result of quasi-static indentation at the indented and rear surfaces, respectively. During indentation, the failure began with a dent on the top side, followed by the commencement and propagation of a fracture on the bottom side. The fracture length on the back surface increased proportionally with indentation displacement. The in-plane extensorial force, which is anisotropic, affects the fracture behaviour of the laminates [63,64]. Additionally, the bottom-side crack was more severe than the top-side crack, indicating that the rear surface sustained more damage and distortion during indentation. This situation was made abundantly evident by the distortion of the PCCP. CPPC demonstrated less distortion because of the presence of carbon in the exterior layer. According to Sezgin and Berkalp [65], strong bonding at the interface and strong adhesion between the carbon ply and matrix improved the mechanical strength of the laminates. The same observation was reported for impact damage of hemp/carbon hybrid laminates, which was studied by Pinto. R et al. [66]. In the study, the c-scan images showed that the hemp layers at the midplane do not affect the laminate’s elasticity at low impact energy levels. When the critical load is reached, the hemp layers’ presence changes the material’s response, resulting in larger damage growth at both the carbon and hemp interfaces based on c-scan images.

Table 6 summarises the damaged areas of the PCCP laminate at two varying ply orientations: [0°/90°_2_, ±45°_2_]_s_ and [±45°_2_, 0°/90°_2_]_s_. Both laminates exhibited the same impact level of indentation (maximum), penetration, and perforation, at 15, 17.5, and 20 J, respectively. The fracture pattern was also identical at each impact level; however, the direction of crack propagation varied due to different ply orientations. In general, the energy in a composite laminate can be easily transferred from one layer to the next if both layers have the same ply orientation, resulting in a higher rate of damage and fracture propagation than if the composite layers had different ply orientations [22]. Consequently, a laminate with varied ply orientations limits energy transfer over its thickness and fails when subjected to higher loads [60]. The interlinear interface between laminates with different ply orientations is weak [61]. Therefore, delamination occurred at the interfaces of the varyingly oriented PALF and carbon layers. Upon impact, local separation from one another causes frequent damage to such systems [59,67].

Based on observation for Table 4, Table 5 and Table 6, the laminates showed four typical main mechanisms of failure due to low-velocity impact, which were caused by the heterogeneous and anisotropic character of fibre reinforced plastic (FRP) laminates: matrix failure, delamination, fibre failure, and penetration [68]. The primary mechanism of internal damage at the rebounding stage of failure is matrix cracking. The cracking is caused by tension, compression, or shear and happens parallel to the fibres. Typically, this manifests as matrix breaking and bonding between the fibre and the matrix [69]. The crack density depends on the degree of mismatch in the properties between the matrix and the fibre, either due to fibre material or orientation [70]. The density matrix cracking was predicted to be higher at interface bonding between carbon fibre and matrix than PALF and matrix. When the applied impact energy is beyond a certain level, the matrix cracking reaches the maximum level, exceeding energy causing the second mode of internal damage mechanism, delamination [71]. Delamination is a fracture that occurs between plies with various fibre orientations in the matrix-rich region [72]. The bending mismatch coefficient between two adjacent laminates, different fibre orientations, and stacking sequences between the layers causes delamination; the delamination area increases as the mismatch coefficient increases [73]. The unstable propagation of delaminations correlates to fast delamination propagation with decreasing force, which is the point at which the maximum compressive force is reached, as shown in Figure 3, Figure 6 and Figure 9, and Figure 12 [74]. Matrix cracking and delamination typically occur earlier in the fracture process than fibre failure damage [75]. High bending stresses and locally high strains under the penetrators cause fibre failure, whereas locally high stresses and the indentation effects of shear pressures cause fibre failure on the unaffected face [68]. The catastrophic penetration mode in Table 4, Table 5 and Table 6 was preceded by fibre failure. Penetration and perforation are failures at the macroscopic level when the penetrators completely enter the material at which the failure of the fibres reaches a critical point [76]. The randomly oriented crack patterns align with the laminates’ ply orientation, as shown in Table 4, Table 5 and Table 6.

## 4. Conclusions

In this study, an in-depth analysis of the low-velocity impact behavior of PALF/carbon hybrid laminate composites at various ply orientations and stacking sequences was performed. The findings of this study led to the following conclusions:By assigning a carbon ply as the exterior layer, the laminate could withstand more impact energy levels. This was most likely due to the excellent mechanical characteristics of carbon that improve the fracture propagation resistance of the laminates;A significant improvement in the maximum contact force was observed when the exterior layer was oriented at ±45°. The shearing effect on the interior and exterior layers was determined to be responsible for this phenomenon;Laminates with varying stacking sequences had a lower energy transfer rate and ruptured when subjected to a higher load. The interlinear interface between laminates with various ply orientations is mechanically weak due to mismatches in the bending deformations of the neighbouring plies;The laminates with ply orientations of [0°/90°] and [±45°]_8_ exhibited better energy absorption than those with ply orientations of [±45°_2_, 0°/90°_2_]_s_ and [0°/90°_2_, ±45°_2_]_s_. This situation is further explained by the fact that the energy in a composite laminate can readily transfer from one ply to the next if both have the same stacking sequence. Laminates with different stacking sequences have a lower energy transfer rate and rupture when subjected to a higher load;PPPP-treated laminates exhibited better impact strength than PPPP-untreated laminates. The elimination of hemicellulose, lignin, waxes, and other contaminants in pure PALF improved the fibre–matrix interaction and increased the impact strength;Delamination occurs at the interfaces of differently oriented PALF and carbon layers, and the prevalent type of damage in such buildings is local separation from one another during impact. The crack propagation was substantially slower in laminates with exterior carbon layers than in laminates with PALF exterior layers.

PALF/carbon hybrid laminate composites have the potential to replace synthetic fibres due to their good mechanical qualities. However, because of their superior characteristics, synthetic fibres cannot be completely replaced.

## Figures and Tables

**Figure 1 materials-15-06121-f001:**
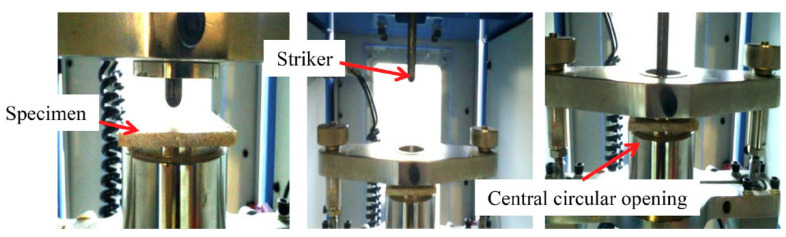
Specimen held and clamped during the drop impact tests.

**Figure 2 materials-15-06121-f002:**
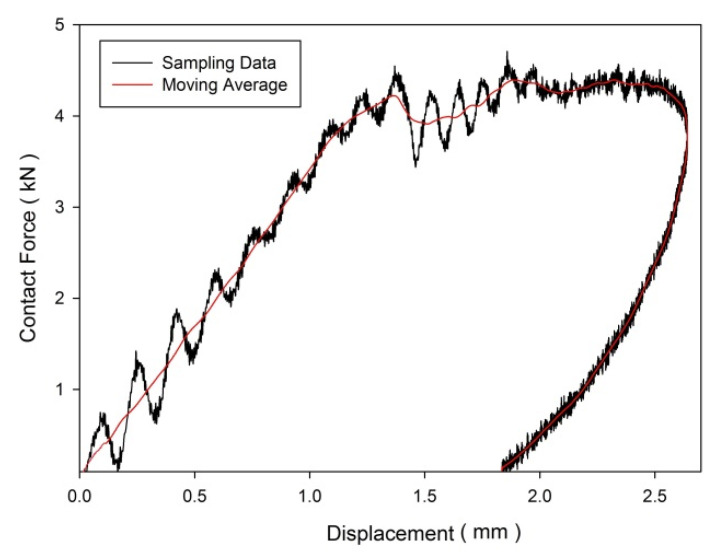
Contact force–displacement of the CPPC laminate at a ply orientation of [±45°]_8_ at an energy level of 10 J.

**Figure 3 materials-15-06121-f003:**
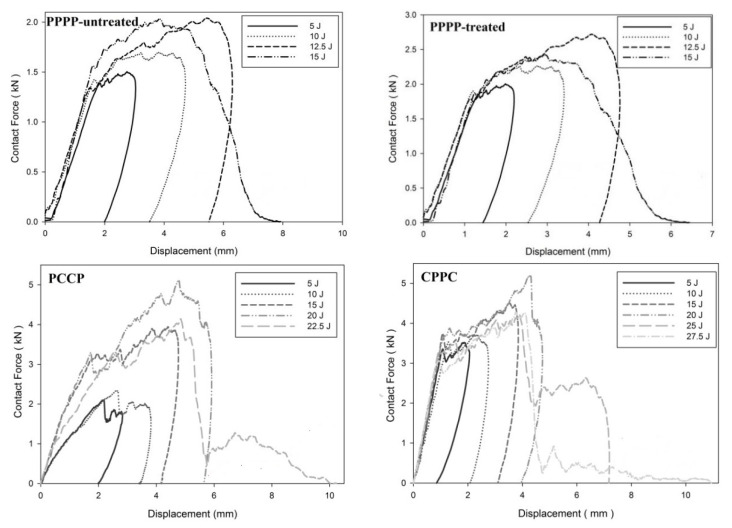
Contact force–displacement of the hybrid laminates at ply orientations of [0°, 90°]_8_.

**Figure 4 materials-15-06121-f004:**
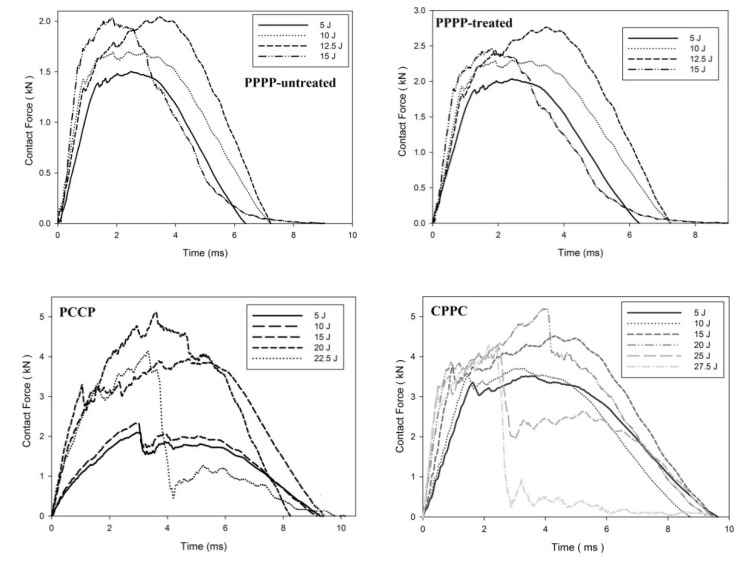
Contact force–time of the hybrid laminates at ply orientations of [0°, 90°]_8_.

**Figure 5 materials-15-06121-f005:**
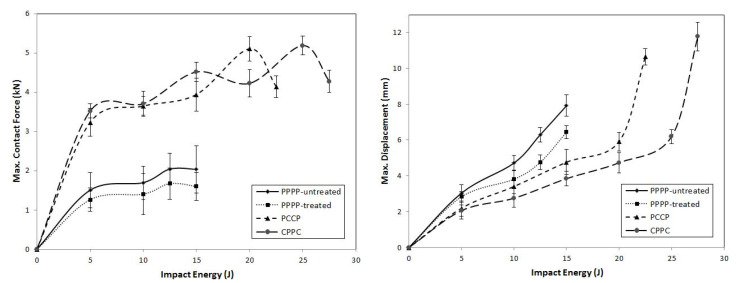
Maximum contact force and displacement against impact energy of the hybrid laminates at ply orientations of [0°/90°]_8_.

**Figure 6 materials-15-06121-f006:**
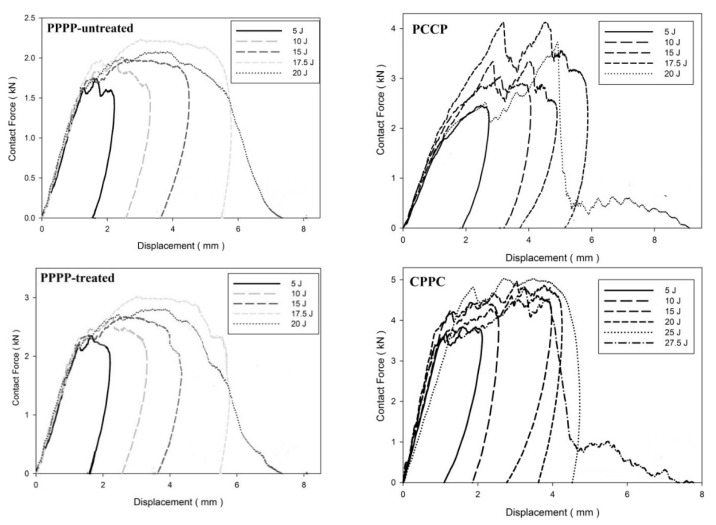
Contact force–displacement of the hybrid laminates at a ply orientation of [±45°]_8_.

**Figure 7 materials-15-06121-f007:**
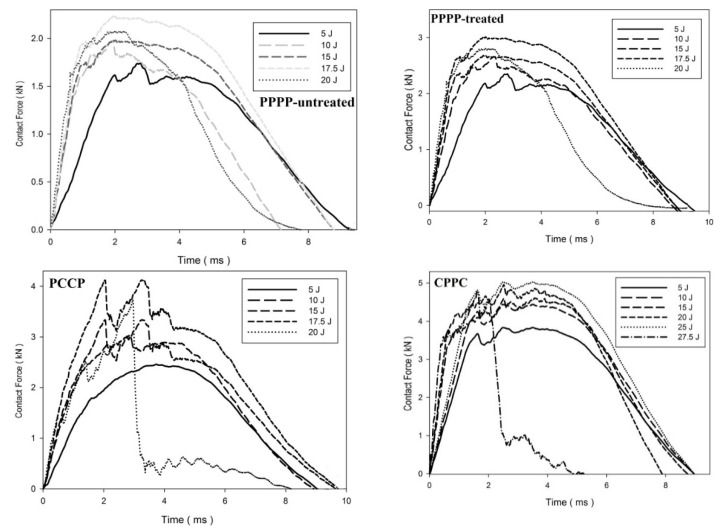
Contact force–time of the hybrid laminates at a ply orientation of [±45°]_8_.

**Figure 8 materials-15-06121-f008:**
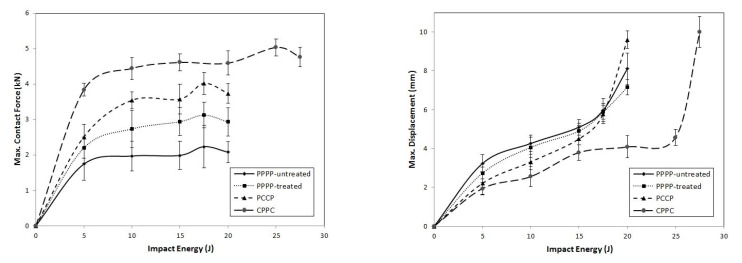
Maximum contact force and displacement against impact energy of the hybrid laminates at a ply orientation of [±45°]_n_.

**Figure 9 materials-15-06121-f009:**
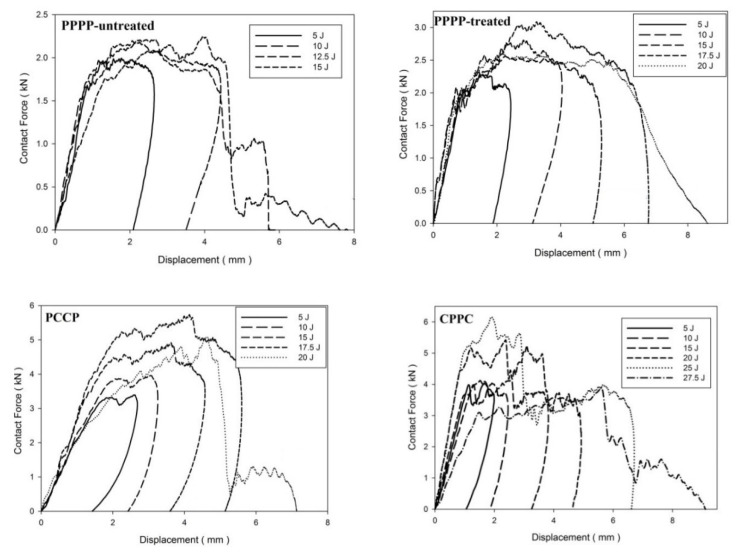
Contact force–displacement of the hybrid laminates at ply orientations of [±45°_2_, 0°/90°_2_]_s_.

**Figure 10 materials-15-06121-f010:**
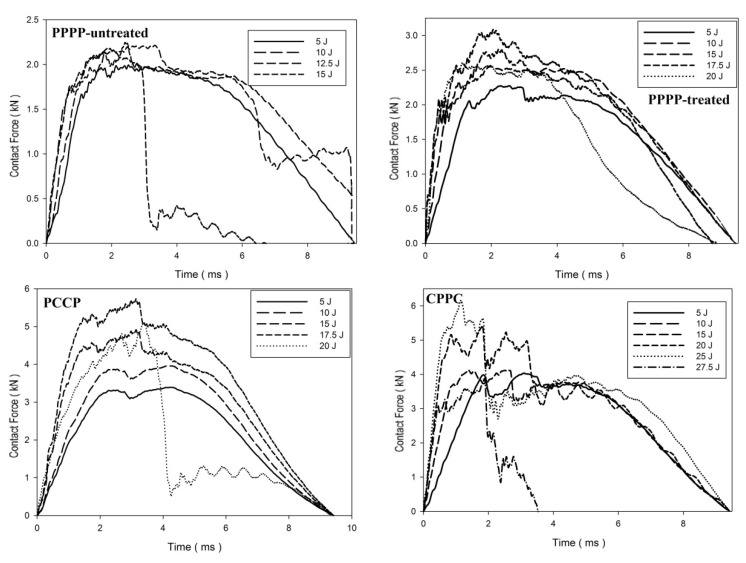
Contact force–time of the hybrid laminates at ply orientations of [±45°_2_, 0°/90°_2_]_s_.

**Figure 11 materials-15-06121-f011:**
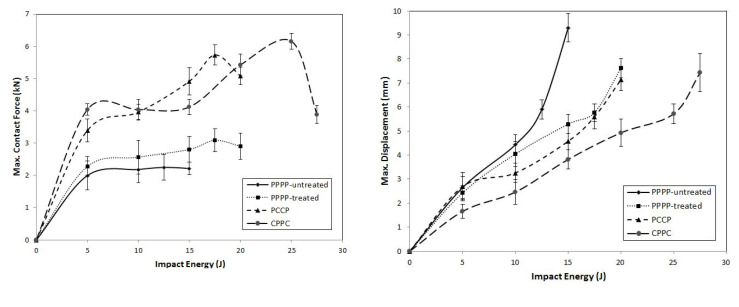
Maximum contact force and displacement against impact energy of the hybrid laminates at ply orientations of [±45°_2_, 0°/90°_2_]_s_.

**Figure 12 materials-15-06121-f012:**
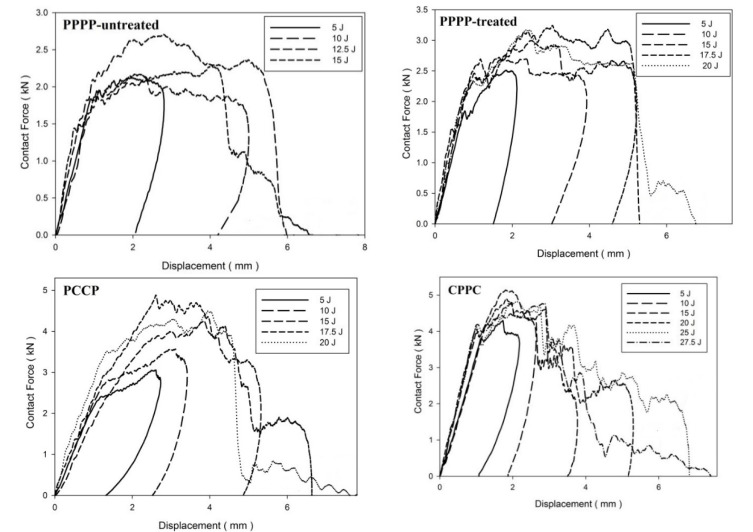
Contact force–displacement of the hybrid laminates at ply orientations of [0°/90°_2_, ±45°_2_]_s_.

**Figure 13 materials-15-06121-f013:**
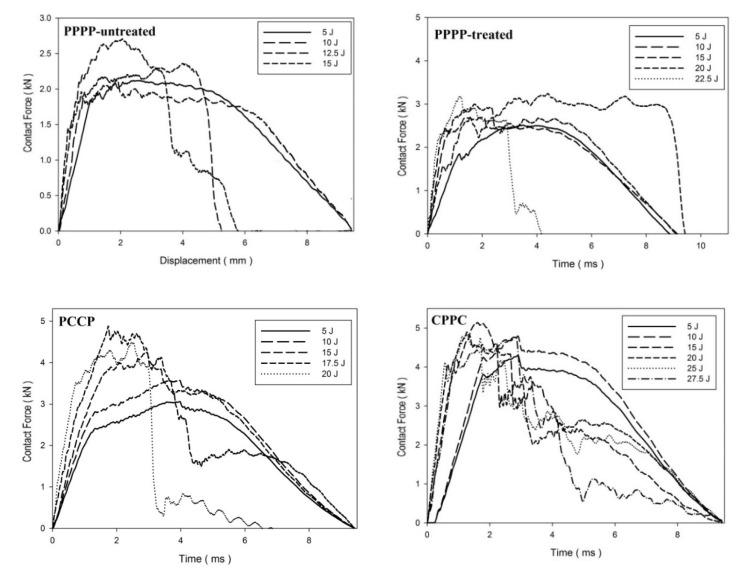
Contact force–time of the hybrid laminates at ply orientations of [0°/90°_2_, ±45°_2_]_s_.

**Figure 14 materials-15-06121-f014:**
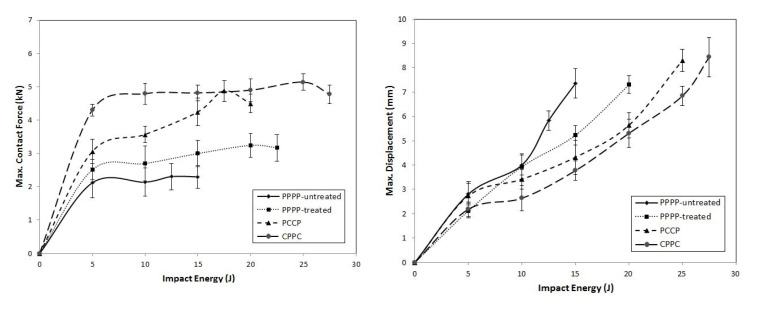
Maximum contact force and displacement against impact energy of the hybrid laminates at ply orientations of [0°/90°_2_, ±45°_2_]_s_.

**Figure 15 materials-15-06121-f015:**
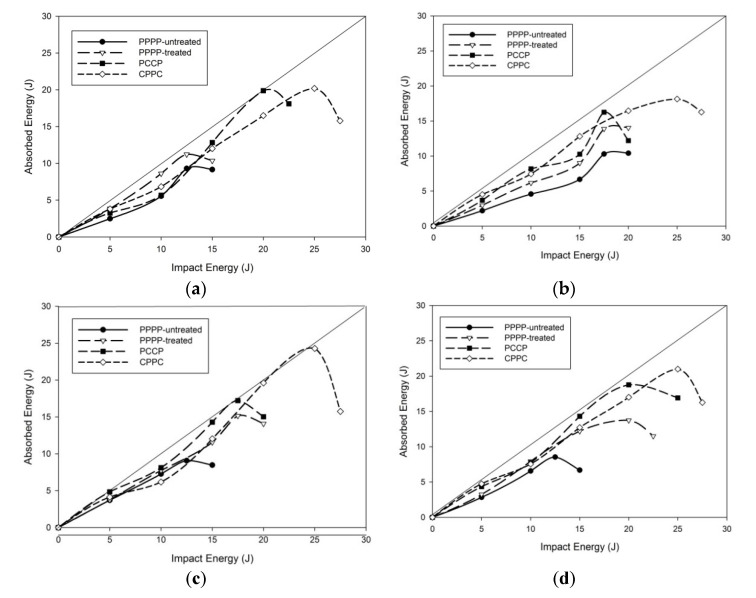
Energy profiling diagram of the hybrid laminates at ply orientations of (**a**) [0°/90°]_8_, (**b**) [±45°]_8_, (**c**) [±45°_2_, 0°/90°_2_ ]_s_, and (**d**) [0°/90°_2_, ±45°_2_]_s_.

**Table 1 materials-15-06121-t001:** Fibres and matrix polymer properties.

Property	Epoxy	Fibre	
		Carbon Fibre	PALF
Tensile Strength (MPa)	55	3530	630
Tensile Modulus (GPa)	1.75	230	10.46
Strain at failure (%)	6	1.5	1.05
Reference	[37]	[38]	[39]

**Table 2 materials-15-06121-t002:** (a) Layering sequence of the laminates: PALF (P) and carbon fibre (C). (b) Illustration of layering sequences.

(a)
Lamination	Orientation	Layering Pattern	Volume Fraction of Fibre (%)	Thickness (mm)
PALF	Carbon Fibre	Total
Cross-ply symmetric	[0°, 90°]_8_	PPPP-untreated	24	-	24	5.83 ± 0.35
PPPP-treated	21	-	21	5.84 ± 0.20
PCCP	16.7	6.2	22.9	5.52 ± 0.11
CPPC	16.7	6.2	22.9	5.44 ± 0.15
Angle-ply symmetric	[±45°]_8_	PPPP-untreated	24	-	24	5.91 ± 0.22
PPPP-treated	21	-	21	5.73 ± 0.24
PCCP	16.7	6.2	22.9	5.55 ± 0.16
CPPC	16.7	6.2	22.9	5.48 ± 0.12
Symmetric Quasi-isotropic	[±45°_2_, 0°/90°_2_]_s_	PPPP-untreated	24	-	24	5.86 ± 0.13
PPPP-treated	21	-	21	5.78 ± 0.22
PCCP	16.7	6.2	22.9	5.53 ± 0.16
CPPC	16.7	6.2	22.9	5.43 ± 0.21
[0°/90°_2_, ±45°_2_]_s_	PPPP-untreated	24	-	24	5.91 ± 0.20
PPPP-treated	21	-	21	5.73 ± 0.18
PCCP	16.7	6.2	22.9	5.60 ± 0.15
CPPC	16.7	6.2	22.9	5.46 ± 0.22
**(b)**
	**PPPP-Untreated**	**PPPP-Treated**	**PCCP**	**CPPC**
[0°, 90°]_8_	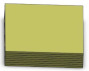	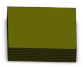	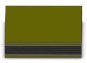	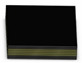
[±45°]_8_	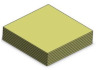	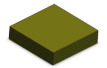	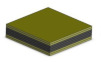	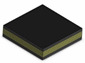
[±45°_2_, 0°/90°_2_]_s_	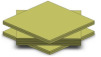	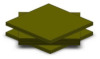	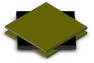	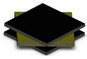
[0°/90°_2_, ±45°_2_]_s_	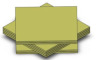	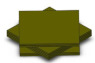	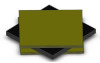	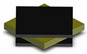

**Table 3 materials-15-06121-t003:** Energy levels used to investigate the impact of the laminates.

Layering Sequence		Energy Impact (J)
[0°, 90°]_8_	PPPP-untreated	5, 10, 12.5, 15
	PPPP-treated	5, 10, 12.5, 15
	PCCP	5, 10, 15, 20, 22.5
	CPPC	5, 10, 15, 20, 25, 27.5
[±45°]_8_	PPPP-untreated	5, 10, 15, 17.5, 20
	PPPP-treated	5, 10, 15, 17.5, 20
	PCCP	5, 10, 15, 17.5, 20
	CPPC	5, 10, 15, 20, 25, 27.5
[±45°_2_, 0°/90°_2_]_s_	PPPP-untreated	5, 10, 12.5, 15
	PPPP-treated	5, 10, 15, 17.5, 20
	PCCP	5, 10, 15, 17.5, 20
	CPPC	5, 10, 15, 20, 25, 27.5
[0°/90°_2_, ±45°_2_]_s_	PPPP-untreated	5, 10, 12.5, 15
	PPPP-treated	5, 10, 15, 17.5, 20
	PCCP	5, 10, 15, 17.5, 20
	CPPC	5, 10, 15, 20, 25, 27.5

**Table 4 materials-15-06121-t004:** Damaged areas of the PPPP-untreated and PPPP-treated laminates at ply orientations of [0°/90°]_8_.

Impact Level	PPPP-Untreated	PPPP-Treated
Top-Side	Bottom-Side	Top-Side	Bottom-Side
Indentation (maximum)	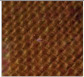	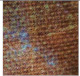	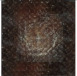	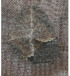
Penetration	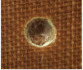	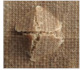	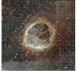	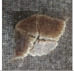
Perforation	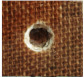	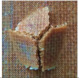	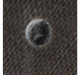	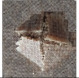

**Table 5 materials-15-06121-t005:** Damaged areas of the PCCP and CPPC at a ply orientation of [0°/90°]_8_.

Impact Level	PCCP	CPPC
Top-Side	Bottom-Side	Top-Side	Bottom-Side
Indentation (maximum)	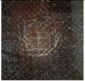	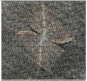	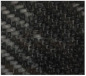	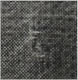
Penetration	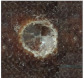	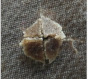	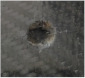	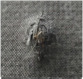
Perforation	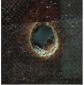	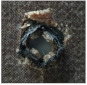	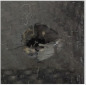	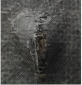

**Table 6 materials-15-06121-t006:** Damaged areas of the PCCP at ply orientations of [0°/90°_2_, ±45°_2_]_s_ and [±45°_2_, 0°/90°_2_]_s_.

ImpactLevel	[0°/90°_2_, ±45°_2_]_s_	[±45°_2_, 0°/90°_2_]_s_
Top-Side	Bottom-Side	Top-Side	Bottom-Side
Indentation (maximum)	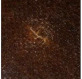	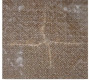	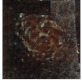	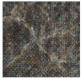
Penetration	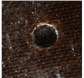	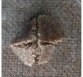	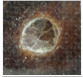	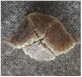
Perforation	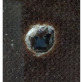	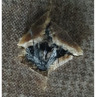	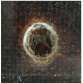	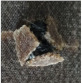

## Data Availability

The raw/processed data required to reproduce these findings cannot be shared at this time as the data also form part of an ongoing study.

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
