# Peer review of "Effects of Ply Orientations and Stacking Sequences on Impact Response of Pineapple Leaf Fibre (PALF)/Carbon Hybrid Laminate Composites"

_materials, 2022, doi:10.3390/ma15176121_

Round 1

Reviewer 1 Report

Top of Form

T

The authors investigate the impact response behaviors of pineapple leaf fibre (PALF)/carbon hybrid laminate composites for different ply orientations and stacking sequences. The laminates were manufactured using a vacuum infusion approach with various stacking sequences and ply orientations classified as symmetric quasi-isotropic, angle-ply symmetric, and cross-ply symmetric. The laminates were analyzed using an IMATEK IM10 drop weight impact tester with an increment of 5 J until the samples were perforated. This investigation reveals that the overall impact properties of PALF and carbon as reinforcements were improved by a beneficial hybridsed effect. The laminates with an exterior carbon layer can withstand high impact energy levels. The laminate with different stacking sequences had a lower energy transfer rate and ruptured at a higher impact energy.

The paper is well-written and could be published after major revision.

The authors need to interpret the meanings of the variables.

Please highlight your contributions in introduction.

What is the blue lines in Fig 2.?

“Error! Reference source not found. lists the damaged areas of the PCCP and”, see line 396.

“Error! Reference source not found. summarizes the damaged areas of the”, see line 417.

“Error! Reference source not found. (c) and (d) illustrate the energy profiling”, see line 346.

The introduction should be supported by papers published by MDPI such as

Bistable Morphing Composites for Energy-Harvesting Applications

Effect of surface preparation on the strength of vibration welded butt joint made from PBT composite

Optimization of abrasive water jet machining of SiC reinforced aluminum alloy based metal matrix composites using Taguchi–DEAR technique

Investigation of mechanical properties of dual-fiber reinforcement in polymer composite

These publications present the integration between ANN and metaheuristic optimizers which may support the introduction section. Modeling of drilling process of GFRP composite using a hybrid random vector functional link network/parasitism-predation algorithm

The abstract should be rewritten to reflect the significance of the proposed work. The current abstract shows a lot of background information.

Conclusion: What are the advantages and disadvantages of this study compared to the existing studies in this area?

The inspiration of your work must further be highlighted. Some suggested recent literatures should add. For example, Recent progresses in wood-plastic composites: Pre-processing treatments, manufacturing techniques, recyclability and eco-friendly assessment.

“The effect of flax fiber (FF) and recycled rubber (RR) on the thermal stability, phase morphology and physico-mechanical properties (tensile, flexion, impact, hardness and density) of hybrid composites were investigated using recycled high-density polyethylene (rHDPE) as the matrix for totalconcentrations of up to 80 wt.%.” this paragraph is not clear for me.

 What are the advantages and limitations of injection and compression from authors point of view?

Author Response

Please refer to the file attached.

Reviewer 2 Report

The authors submitted a manuscript on the low velocity impact behaviour of PALF/carbon hybrid composites, evaluating the effect of different stacking sequences. The manuscript reports some interesting results, but it needs a significant revision before it can be considered for publication. More specific comments are as follows:

1.       The quality of English language needs significant improvements throughout the text (please also choose between America or British English). There are also many misprints (for instance, the last author name is badly written, D. Hui). Figures in the pds file are not correctly identified (Error, reference source not found)

2.       Abstract: the authors should include some key quantitative results, highlighting the differences among configurations;

3.       Line 28: GFRPs is not the acronym of carbon fibre reinforced polymers;

4.       Introduction: lines 51-70, This paragraph might be eliminated, as it provides well known information. In addition, the meaning of “delamination occurs because of the vertical displacement…..fibers” is not clear;

5.       Lines 89-93: what is the reference?

6.       Introduction should be more focused on providing an in-depth state of the art and review of composites based on carbon/PALF composites, emphasizing why this combination might be suitable for impact energy absorption applications;

7.       What do the authors mean by “twill plain weave”? What is the fibre areal weight of PALF fabric?

8.       Table 1: Are these properties belonging to the same fibres and matrix used in the present study? If not, this table is useless;

9.       Ref [29] does not seem to be pertinent;

1.   The actual thickness of all the configurations must be included. It is not clear how the fibre volume fraction included in table 2 was obtained. Was it measured or estimated? The stacking sequences should be better described in terms of the different fibres, as it is not clear in the quasi-isotropic configurations, which layers were PALF and which ones were Carbon. A picture is required;

1 ASTD D7136 does not suggest the specimen’s dimension of 70x70 mm (it should be 150x100 mm);

1  What is the diameter of the central aperture?

1As a general comment, what is lacking in this manuscript is a sound discussion of the damage mechanisms. There is no analysis of the internal damage modes (no C-scans, no cross-section observations) and no analysis of the residual indentation depth. There are some hypotheses, taken from the literature, but not experimentally verified. In many cases, the explanations are not clear, such as in lines 197-198, 207 (identically different?), 229-231, 341-344, 351-352 (carbon fibres are more brittle than PALF fibres?), 388-389 (the fracture patterns appear to be the same), 412-414;

1The authors should include the Energy vs time curves, maybe as supplementary info, because in some cases there is evidence of rebound but the authors indicated penetration;

1  Data should be provided with standard deviation;

1 As regards the energy profile diagrams, they should have the same scale in X and Y axes, and the equal energy line should be included in the plots;

1 Front and back surfaces of impacted specimens should be compared as a function of impact energy, to show the evolution of damage. In table 6, the pictures of configuration [0/90,+-45] are the same that have been included in table 5 for [0/90] configuration, while in table 4, for PPPP-treated at penetration, the pictures seem similar to the ones in table 6 central line (just rotated and magnified). The other samples are missing.

Author Response

Please refer to the file attached

Round 2

Reviewer 1 Report

Accept .

Author Response

The manuscript has been proofread accordingly. 

The manuscript has been accepted for publication by reviewer 1.

Reviewer 2 Report

The authors revised the manuscript but the main issues are still in place and have not been properly addressed.

1)    There are still instances of British and American English;

2)    Introduction has not been improved, and no state of the art or review of previous works dealing with carbon and PALF fibres have been provided;

3)    Thickness of the laminates is of fundamental importance, especially for impact tests. Therefore, simply adding a range spanning from 5 to 6 mm is not sound. The actual thickness of each laminate should be added. In addition, no answer has been provided about fibre volume fractions;

4)    If the impact tests were performed according to ASTM D7136, the fixture used (with a central aperture) does not seem to be the correct one (The cut-out in the plate shall be 75 x 125 mm);

5)    The discussion of internal damage is missing. The authors state that this is out of paper’s scope, but they are comparing the impact resistance of different stacking sequences, therefore it cannot be ignored;

6)    The authors did not include the Energy vs Time curves (they were suggested as supplementary material), and they did not answer the question about “in some cases there is evidence of rebound but the authors indicated penetration”;

7)    Most data were presented without standard deviation. Data in figs. 6, 8, 10, for instance, are for given parameters (force, displacement) and must be provided with the corresponding standard deviation to appreciate differences;

8)    In the energy profile diagrams, for better readability, both axes should start from 0 and end at 30, and the equal energy line should be added;

9)    The authors stated that the properties listed in table 1 come from exactly the same materials as those used in the present work. This is not correct, as for instance ref 29 is related to a polyester resin and not to an epoxy resin. In addition, PALF mechanical properties are for a specific PALF cultivar tested in 2009, and it is unlikely that the mechanical properties can be considered the same.

Author Response

Kindly refer attachment.
